# Tunable Properties *via* Composition Modulations of Poly(vinyl alcohol)/Xanthan Gum/Oxalic Acid Hydrogels

**DOI:** 10.3390/ma15072657

**Published:** 2022-04-04

**Authors:** Alin Alexandru Enache, Diana Serbezeanu, Tăchiță Vlad-Bubulac, Alina-Mirela Ipate, Dana Mihaela Suflet, Mioara Drobotă, Mihaela Barbălată-Mândru, Radu Mihail Udrea, Cristina Mihaela Rîmbu

**Affiliations:** 1Apel Laser, Street Vanatorilor 25, Mogosoaia, 077135 Ilfov, Romania; alin.enache@apellaser.ro (A.A.E.); radu.udrea@apellaser.ro (R.M.U.); 2“Petru Poni” Institute of Macromolecular Chemistry, 41A Aleea Gr. Ghica Voda, 700487 Iasi, Romania; tvladb@icmpp.ro (T.V.-B.); alina.ipate@icmpp.ro (A.-M.I.); dsuflet@icmpp.ro (D.M.S.); miamiara@icmpp.ro (M.D.); mandru.mihaela@icmpp.ro (M.B.-M.); 3Department of Public Health, Faculty of Veterinary Medicine, Iasi University of Life Sciences (IULS), Mihail Sadoveanu Alley no. 8, 700490 Iasi, Romania

**Keywords:** hydrogels, poly(vinyl alcohol), xanthan gum, swelling behavior, mechanical properties, antimicrobial activity

## Abstract

The design of hydrogel networks with tuned properties is essential for new innovative biomedical materials. Herein, poly(vinyl alcohol) and xanthan gum were used to develop hydrogels by the freeze/thaw cycles method in the presence of oxalic acid as a crosslinker. The structure and morphology of the obtained hydrogels were investigated by means of scanning electron microscopy (SEM), Fourier transform infrared (FTIR) spectroscopy, and swelling behavior. The SEM analysis revealed that the surface morphology was mostly affected by the blending ratio between the two components, namely, poly(vinyl alcohol) and xanthan gum. From the swelling study, it was observed that the presence of oxalic acid influenced the hydrophilicity of blends. The hydrogels based on poly(vinyl alcohol) without xanthan gum led to structures with a smaller pore diameter, a lower swelling degree in pH 7.4 buffer solution, and a higher elastic modulus. The antimicrobial activity of the prepared hydrogels was tested and the results showed that the hydrogels conferred antibacterial activity against Gram positive bacteria (*Staphylococcus aureus 25923 ATCC*) and Gram negative bacteria (*Escherichia coli 25922 ATCC*).

## 1. Introduction

Hydrogels are well-known soft materials, three-dimensional designed polymeric networks that have the capability to retain a large amount of water without losing their structural integrity [1]. Holding a variety of adaptable chemical, physical, and biological properties, hydrogels have been extensively studied as versatile materials for specific applications [1,2,3]. A biocompatible, transparent, malleable, bioinert polymer, commonly used as a hydrogel in pharmaceutical applications is poly(vinyl-alcohol) (PVA) [4,5,6] (Figure 1). PVA-based hydrogels have a high resistance to solvents, oils, and greases, as well as a higher resistance to oxygen permeability compared to other polymers [7,8,9]. Moreover, PVA is a water-soluble semicrystalline polymer with a higher content of hydroxyl groups, groups that can be easily induced to form a stable hydrogen bonding network. PVA, as a synthetic polymer, can be crosslinked [10] both by the freeze/thaw technique and by a variety of other methods such as chemical crosslinking [4,11,12,13] or irradiation [14]. Crosslinking is a useful tool in modifying the properties of existing polymers in order to obtain new and improved polymeric materials. Modification of PVA by its hydroxyl groups opens up new fields of useful applications of this material [15]. However, the main disadvantages of PVA-based hydrogels are their poor mechanical properties and their slow response to swelling [16]. To improve these characteristics, multicomponent networks capable of increasing the viscosity of the solution have been designed, even at low concentrations such as polysaccharides that have a high affinity for water [17]. Xanthan gum (XG) is generally considered a non-gelling polymer [10] but it can induce synergistic effects with many biocompatible polymers, resulting in a higher viscosity in the mixture or in the formation of a gel. Its macromolecules (Figure 1) are rich in hydroxyl, carboxyl, and other functional groups, resulting in good adsorption properties and a strong ability to coordinate with heavy metal ions [18,19]. The addition of xanthan gum to PVA elastomeric material has an important influence on its properties in terms of structural compatibility and polymer morphology [10,20,21,22,23].

Numerous studies describe the compatibility of mixtures of polymeric materials, but there are very few studies on the preparation and characterization of mixtures of biodegradable polymers. Therefore, mixing PVA and xanthan gum in a specific ratio under certain conditions (temperature) can lead to composite hydrogels with excellent combined properties. For example, Ray and co-authors [22] developed XG-PVA pH-sensitive interpenetrating network microspheres by the emulsion crosslinking method using glutaraldehyde as crosslinking agent. In their study, the authors showed that the amount of crosslinking agent and XG content influences the drug entrapment efficiency and its release from microspheres. However, the use of glutaraldehyde as a crosslinker remains problematic due to its toxicity, which may limit certain applications of the developed microspheres. Other authors discussed the interaction between PVA and XG components in several XG/PVA mixtures using different techniques [24]. Wei and others [21] prepared an environmentally friendly dust suppressant interpenetrating double-network hydrogel starting from PVA, XG, and acrylic acid. The authors studied the effect of composition and the influence of reaction temperature on the viscosity, surface tension, and compression strength in order to obtain hydrogels with optimal properties for the targeted application [21].

Considering all these facts, the aim of our study was to use a non-toxic crosslinking agent, namely acid oxalic, in order to control and enhance the hydrolitic stability and the mechanical properties of PVA/XG-based hydrogels. Thus, by composition modulation, various ratios of poly(vinyl alcohol) and xanthan gum were dissolved in a certain amount of double-distilled water and homogenized at room temperature with continuous stirring, then, the required amount of oxalic acid was introduced in the system in order to obtain new materials with guided properties. The hydrogels obtained via freezing/thawing technique were lyophilized to a dry state and then thermally treated to achieve supplementary physical crosslinking, in addition to the chemical crosslinking in the presence of oxalic acid. The physical, chemical, morphological, mechanical, and antimicrobial activity of the PVA/XG hydrogels were characterized by SEM, FTIR, porosity measurements, swelling behavior, mechanical properties, and antimicrobial assay.

## 2. Materials and Methods

### 2.1. Materials

Poly(vinyl alcohol) (MW 9000~10,000, 87–89% hydrolyzed) and oxalic acid were purchased from Merch (Darmstadt, Germany). Xanthan gum transparent grade was purchased from Elements (Vienna, Austria). All other reagents were used as received from commercial sources or were purified by standard methods.

### 2.2. Methods

A LUMOS microscope Fourier transform infrared (FTIR) spectrophotometer (Bruker Optik GmbH, Ettlingen, Germany) equipped with an attenuated total reflection (ATR) device was used to record the scans of hydrogels between 4000 and 500 cm^−1^ at a resolution of 4 cm^−1^.

Scanning electron microscopy (SEM, FEI Company, Brno, Czech Republic) was performed on a Quanta 200 instrument, at 30 kV, with a magnification of 380–3600. For these investigations, small pieces (≈5 mm) of the PVA/XG hydrogels were attached by conductive adhesive tape to an aluminum specimen holder. Then, the PVA/XG hydrogels were inserted in the SEM microscope and observed in high vacuum using a secondary electron detector at accelerating voltage equal to 30 kV. The diameters of the pore were measured by means of Image J program. At least 25 pores from each sample were taken in consideration to obtain the average pore diameters.

The porosity of the hydrogel based on PVA crosslinked with oxalic acid and xanthan gum was determined by the liquid displacement method. Ethanol has been used as a displacement liquid because it penetrates easily into the pores and does not cause the polymer matrix to shrink or swell. The dehydrated sample (approx. 0.1 g) was placed in a graduated cylinder containing 2 mL (*V*_1_) of ethanol and stored at 37 °C for 6 h. Prior to weighing, the sealed cylinder containing the sample to be investigated was placed in an ultrasonic bath for 10 min to force the ethanol to penetrate the pores of the polymer matrix giving the total volume denoted as *V*_2_. Subsequently, the sample was removed from the cylinder and the volume of remaining ethanol was recorded as *V*_3_. The porosity (*P*) of the polymer matrix was calculated using the following equation:(1)$$P\left(\%\right)=\frac{V_{1}-V_{3}}{V_{2}-V_{3}}\times 100$$

Hydrogel samples based on poly(vinyl alcohol) and xanthan gum crosslinked with oxalic acid in various proportions were used in determining the degree of swelling in the buffer solution (pH 7.4). Dimensions of these hydrogels were chosen as 0.5 × 0.5 × 0.5 cm^3^. The samples were weighed at different times, both dry and wet after being immersed in the buffer solution. At fixed time intervals, the hydrogels were removed from the buffer solution, and the excess solution was removed with filter paper. The experiment ended when no significant change in weight of the hydrogels was measured. The degree of swelling of these hydrogels based on xanthan gum and poly(vinyl alcohol) with oxalic acid was determined using the calculation formula:(2)$$S\left(\%\right)=\frac{W_{t}-W_{d}}{W_{d}}\times 100$$
where *S(%)*—the amount of absorbed buffer solution; *W*_d_—weight of the dry hydrogels; *W*_t_—weight of the hydrated hydrogels. A swelling kinetics model was studied based on the swelling ratio-time curves.

Uniaxial compression tests were performed using a texture analyzer (Brookfield Texture PRO CT3^®^, Brookfield Engineering Laboratories Inc., Middleboro, USA) at room temperature. The hydrogels in the hydrated state (24 h, PBS 7.4) with cylindrical shape (10–12 mm diameter and 6–9 mm height) were compressed with 0.067 N trigger load between two parallel plates with a compression rate of 0.1 mm/s up to 70% strain. The elastic modulus (*E*) was calculated according to Equation (3):(3)$$E=\frac{\sigma }{\varepsilon }=\frac{F/A}{\Delta l/l_{0}}$$
where *σ* is the compressive stress (N/m^2^), *ε* is the strain, *F* is the force (N), *A* is the cross-sectional area of the hydrogel (m^2^), Δ*l* is the change in length (m), and *l*_0_ is the original length (m). The elastic modulus *E* was calculated from the slope of stress-strain curves between 5 and 15% compressions.

The antibacterial activity of the PVA/XG hydrogels was evaluated against Gram positive bacteria (*Staphylococcus aureus 25,923 ATCC*) and Gram negative bacteria (*Escherichia coli 25,922 ATCC*) according to a method adapted from a previously reported work [25].

The antimicrobial activity of PVA/XG hydrogels was qualitatively determined using the Kirby–Bauer diffusimetric method adapted for this type of biomaterial. For this purpose, standardized bacterial cultures *Staphylococcus aureus* ATCC 25,923 (Gram positive) and *Escherichia coli* ATCC 25,922 (Gram negative) were brought to a cell density corresponding to 0.5 McFarland turbidity standard (1.5 × 10^8^ CFU/mL). The microbial suspensions were seeded on the surface of Muller–Hinton agar solid culture medium, after which PVA/XG hydrogel samples (0.05 g) were distributed. After incubation at 37 °C for 24 h, the microbial zone of inhibition formed around the hydrogels was monitored to confirm the antimicrobial activity of the tested samples.

Antimicrobial activity was also tested using the contact time technique (CT), which was adapted to the hydrogels to show their inhibitory effect on the tested strains over time.

For this purpose, each sample of hydrogel (0.05 g) was dispersed in 5 mL of a microbial suspension *Staphylococcus aureus* ATCC 25,923 and *Escherichia coli* ATCC 25922. To determine the inhibitory effect of the hydrogel samples, 3 contact times were established: t0, t1 (24 h/37 °C), and t2 (48 h/37 °C). The technique was to disperse and introduce 1 mL of each microbial suspension in 20 mL of molten Muller–Hinton culture medium cooled to 45 °C, which was in contact with the hydrogel for a predetermined time. After homogenizing the suspensions and solidifying the culture medium in Petri dishes, the samples were incubated at 37 °C. After every 24 h of incubation, the number of colony-forming units (CFU/mL test suspension) was determined (Figure 2).

### 2.3. Preparation of PVA/XG Hydrogels

PVA-xanthan gum hydrogels crosslinked with oxalic acid were prepared using PVA and xanthan gum as polymer networks and oxalic acid as crosslinker by the freezing-thawing method followed by thermal treatment. Various mass ratios of PVA and xanthan gum were dissolved in an appropriate amount of double-distilled water and homogenized with continuous stirring at room temperature, after which the required amount of oxalic acid was introduced and stirring was continued until a homogeneous mixture was obtained. Subsequently, the resulting mixture was freeze/thawed for 5 days. Drying of these new polymeric materials was performed by lyophilization. The hydrogels were introduced into the oven and kept at 100 °C for 1 h. A detailed example of the preparation of hydrogels is described for the sample denoted as PVA/XG-100/0:0.5 g PVA was dissolved in 7.25 mL in deionized water at 90 °C for 6 h. At the cooled solution of PVA, 0.08 g oxalic acid was added under stirring. The obtained solution was poured into a Petri dish. The solution was subjected to the freeze/thawed cycles for 5 days (−15 °C for 24 h and 1 h at room temperature). A thermal treatment at 100 °C for 1 h was applied. Table 1 shows the compositions expressed in mass ratios, as well as the codes of the new polymer matrices.

## 3. Results and Discussions

### 3.1. Structural Characterization

The crosslinking of PVA via esterification reaction in the presence of oxalic acid and the formation of PVA-XG networks were monitored by attenuated total reflection-Fourier transform infrared (ATR-FTIR) spectroscopy. ATR-FTIR spectra of pure PVA, XG, and their PVA/XG networks are shown in Figure 3a. The spectra of pure PVA and pure XG reveal the presence of strong hydrogen bonds due to the intra and intermolecular interaction of hydroxyl groups which are associated with the broad characteristic bands located in the interval of 3600–3100 cm^−1^ (OH stretching vibration). In ATR-FTIR spectrum of PVA, the band observed at 2929–2915 cm^−1^ corresponds to symmetrical ν_CH2_ vibrations, the band observed at 2854 cm^−1^ corresponds to asymmetrical ν_CH2_, while the band observed at 1245 cm^−1^ is related to ν_C–O_ vibration. The peak at 1430 cm^−1^ is assigned to the vibration ω_CH2_, while the peaks located at 1375, 1327, 1092, and 847 cm^−1^ are assigned to the δ_CH2_ vibrations, C–O and O–H bending, respectively [26,27]. ATR-FTIR spectrum of XG revealed characteristic bands located at 1575, 1425, and 615 cm^−1^ due to the presence of COO groups (C–O asymmetric stretching and C–O symmetric stretching), along other bands at 2939 cm^−1^ and 1200–990 cm^−1^ that are common to all polysaccharides, representing O–H bonds and C–H bonds of CH_2_ groups located on the glucose rings [28].

The ATR-FTIR spectra of PVA/XG hydrogels revealed characteristic absorption peaks of both PVA and XG indicating their contribution in the resulting samples. As shown in Figure 3a (blue curve), the spectrum of PVA/XG-100/0 sample revealed characteristic absorption bands of the crosslinked PVA (PVA-oxalic acid). Thus, the absorption band at 1730 cm^−1^, which is absent in the spectrum of pure PVA (Figure 3b (black curve)), is due to C=O stretching vibration of ester groups which appeared as a consequence of the crosslinking of PVA via the esterification reaction in the presence of oxalic acid. Moreover, a noticeable change in the alure and a decreased intensity of the peaks located in the region of 3600 to 3100 cm^−1^ indicates that the esterification between hydroxyl groups of PVA and carboxyl groups of oxalic acid took place. The crosslinking mechanism is presented in the Figure 3 inset. The PVA/XG hydrogels exhibited characteristic absorption bands in the region 3600–3000 cm^−1^, which are attributed to the stretching vibration of -OH. The absorption bands centered at 2929–1915 and 2854 cm^−1^ are due to the symmetrical and asymmetrical νCH_2_ vibration. The disappearance of the 1575 cm^−1^ absorption band in the PVA/XG hydrogels compared to the pure compounds can be attributed to the intermolecular hydrogen bonding between XG and PVA [29]. Furthermore, it was observed that the peaks located at 1645 cm^−1^, characteristic for νC=O vibration of pyruvate, and the peak at 1733 cm^−1^ corresponding to νC=O of carboxyl increased in the intensity, as the XG content increased in the final PVA/XG hydrogels.

### 3.2. Morphological Investigation

The section morphology of the microstructured PVA-XG hydrogels chemically crosslinked with oxalic acid was analyzed using SEM. Representative images of the hydrogel samples, lyophilized then heated for 1 h at 100 °C, were taken from sample series and are presented in Figure 4. As it can be seen, the polymeric PVA/XG networks showed porous structures, the pore diameter being dependent on the content of xanthan gum in the hydrogel composition. Thus, the SEM images for PVA/XG-100/0 show parallel, uniform longitudinal pores, continuously organized in a dense matrix that formed a layer of uniform thickness. Starting with the addition of XG (samples PVA/XG-90/10, PVA/XG-80/20, and PVA/XG-70/30), microporous structures with interconnected pores were revealed, indicating the interpenetrating nature of the interactions between polymer chains belonging to different polymer classes. The successful incorporation of the polysaccharide (XG) into the PVA macromolecular chain chemically crosslinked with oxalic acid resulted in interpenetrated network structures with uniform pores that increased in size as the content of XG was increased. Meanwhile, for the PVA/XG-60/40 and PVA/XG-50/50, SEM images show that the pores are round and evenly distributed and also that the pores are interconnected. The presence of the interconnected pores is very important for the practical application because it is easy to encapsulate the bioactive materials.

### 3.3. Porosity

The porosity of the material is a property of solids that determines the size and number of pores inside the material and describes their distribution in the analyzed area. Porosity is defined as the ratio of the volume occupied by the pores to the total volume of the material and is given as a percentage. It depends on the synthesis parameters, as well as on the chemical structure of the compounds used. The inner open pores are interconnected to the surface of the material. On the other hand, closed pores are not connected to each other, reducing the actual density of the material, and influencing biochemical and cytological interactions in the body [30,31,32]. They also do not increase the active surface of the material. The porous surface of the implants affects the osseointegration process [33,34]. Moreover, porous biomaterials can act as drug carriers because they allow the placement of a drug substance in the pores of the material [35,36]. Porous biomaterials must meet the requirements for open and total porosity. However, in addition to determining the porosity, it is also important to characterize the geometric dimensions of the pores. The size and types of pores determine the connection of the implant with the tissue by its growth in the biomaterial pores. Moreover, the control of the parameters allows the direct control of the amount of drug that can be introduced into the biomaterial and delivered to the body. Thus, taking into account all the aspects presented above, porosity was investigated in the case of PVA/XG hydrogels. In Table 2, the most important data regarding the porosity of the investigated samples are listed.

From Table 2, it can be seen that the porosity of PVA/XG hydrogels varies in the range of 67–86%. It has been established that porosity is a key factor when designing and preparing soft materials, such as hydrogels, for biomedical applications. A higher porosity of the biomaterial is responsible for a series of biological activities such as fluids exchange, cells migration, permeability, protein absorption, etc. In our study, the PVA/XG-50/50 sample having 86% porosity could be considered of interest for further biomedical investigations. In addition, from Table 2 we can see that the rated PVA/XG-60/40 sample has a porosity of only 67%. The SEM images are consistent with the data obtained from the porosity investigation (Figure 5). The average pore diameter, calculated using the ImageJ program, was in the range 0.028 ± 0.007–0.064 ± 0.024 mm (Table 2).

### 3.4. Swelling Behavior and Kinetic Studies of PVA/XG Hydrogels

The swelling behavior, as the main characteristic of hydrogels, depends on the nature of the polymers/components (nature of the charges, ionization capacity of the hydrogel depending on pH, permeability of molecules in solution, crosslinking density, etc.), and the conditions under which the experiments are conducted (pH and ambient temperature) [37,38,39]. The swelling process takes place by properly combining the polymer matrices and the solvent. The polymer network comes into contact with the solvent and begins to swell due to the thermodynamic compatibility between the polymer chains and the solvent. Thus, the inflation forces are counterbalanced by the retraction forces induced by the crosslinks in the system, the inflation balance being reached when these forces become equal. In the case of ionic systems, the degree of swelling, S, is determined by the electrostatic repulsions between the macromolecules due to the presence of charges in the structure of the polymer [40]. The chemical structure of the components of PVA systems stabilized with oxalic acid and xanthan gum influenced the values obtained for parameter S. From a practical point of view, the determination of this parameter is extremely important, being closely related to the processes of incorporation and release of active principles. Figure 6 shows the swelling profiles of hydrogels based on PVA stabilized with oxalic acid and xanthan gum in a phosphate buffer solution of pH 7.4, at 37 °C. From Figure 6, it can be observed that the swelling kinetics follow a similar trend. A quick increase could be observed during the initial time periods after immersion of the PVA/XG hydrogels in the buffer solution, followed by a slower swelling process occurring until the equilibrium swelling capacity was reached. Balance was reached after 60 min for all samples. The equilibrium swelling capacity, S_eq_, took high values for each sample and increased in most cases with increasing xanthan gum content (Figure 6, Table 2).

From Table 2, we can observe that the addition of XG to the PVA matrix increases the swelling ratio of hydrogels. Addition of XG (PVA/XG-80/20 and PVA/XG-70/30) leads to an increase in the swelling degree. Further increase in XG concentration (PVA/XG-60/40 and PVA/XG-50/50) yielded further decrease in swelling degree if we compare with the samples denoted PVA/XG-70/30. With the introduction of xanthan gum into the PVA/XG-100/0 matrix, the properties of the membrane were improved, leading to greater swelling in the phosphate buffer pH 7.4 solution and different pore sizes or maintaining a humid environment.

In order to investigate the swelling kinetics of the PVA/XG hydrogels, the experimental swelling data were assessed by Schott’s second-order kinetic model, which tests the experimental data for the entire period of swelling [41]. The equation of the swelling kinetics model is presented in the form (Equation (4)):(4)$$\frac{dS}{dt}=k_{s}{\left(S_{\infty },-,S\right)}^{2}$$

In this equation, k_s_ is the constant rate of swelling, and S and S_∞_ describe the swelling capacity at time t and the theoretical equilibrium buffer solution absorption capacity. After an integration between the limits t, Equation (4) becomes:(5)t/s=A+Bt

Finally, the obtained equation describes the constant A, a reciprocal of the initial swelling rate of the hydrogel (A = 1/k S_∞_^2^), and B, the reciprocal of the maximum swelling (B = 1/S_∞_). The linear relationships between t and t/S are represented in the graphs of Figure 7.

The kinetic parameters are presented in Table 2. The regression coefficients (R) obtained are above 0.997, which shows that Schott’s kinetic equation was adequate to describe the whole swelling process. Moreover, the values of S_∞_ for the PVA/XG hydrogels are characterized by an increase of the values together with the increase of the ratio of XG in the PVA matrix. In addition, the values of the constant k are different, thus suggesting different degrees of swelling of the hydrogels obtained on the basis of poly(vinyl alcohol) and xanthan gum with oxalic acid.

### 3.5. Mechanical Properties

The mechanical properties of PVA/XG hydrogels are decisive factors for their future applications. Compression tests were performed to get information about the elastic behavior of the PVA/XG hydrogels (Figure 8). The values of elastic modulus were calculated between 5 and 15% compression, where the stress–strain curves are linear (Figure 8, inset), and the obtained values are presented in Figure 9. From Figure 8, it can be observed that compressive stress decreases with increasing XG content, the same results have been presented in the literature by other authors [20].

### 3.6. Antibacterial Activity of PVA/XG Hydrogels

The antibacterial activity of the PVA/XG hydrogels was evaluated using *S. aureus* and *E. coli* (Figure 10). Compared with PVA/XG hydrogels without XG, hydrogels containing XG showed a wide spectrum of antibacterial activity against *S. aureus* and *E. coli.* The results showed that these hydrogels inhibit bacterial infection demonstrated that the hydrogels hold potential for medical devices.

The antimicrobial potential was also demonstrated by the contact time technique. It was found that all hydrogel samples inhibited the multiplication of microbial cultures after 24 h and maintained this for up to 48 h (Figure 11, Table 3).

It has been demonstrated in the scientific community that, in general, several factors are responsible for inducing antibacterial activity in natural polysaccharides. Worthy of being mentioned here are the nature and the content of functional groups which are present in PVA/XG, our hydrogel construct (PVA and XG hydroxyl groups, glucuronic acid, acetate, pyruvate etc.), the content of glucose segments from xanthan gum, morphology, surface area, etc. [42,43]. An inhibitory effect was observed also for the sample PVA/XG-100/0 against *Staphylococcus aureus* and *Escherichia coli*, probably as a result of the presence of hydroxyl groups of the PVA molecules and some residual carboxyl of the chemical crosslinker.

## 4. Conclusions

The morphology and the architectural configuration of the hydrogels based on polyvinyl alcohol/xanthan gum have been controlled in this study by both physical and chemical conditions in the presence of oxalic acid. Studies have shown that oxalic acid has led to new semi-interpenetrating networks. A temperature program was also used for complete crosslinking. The synthesized hydrogels showed microporous structures, according to SEM studies, with pore diameter strictly dependent on the XG concentration in the system. With growth percentage in XG, pore size increased. Swelling studies performed in pH 7.4 phosphate buffer and at 37 °C showed a dependence on the XG content, obtaining higher values of degree of swelling, S, with increasing concentration in polysaccharide units. As insoluble hydrogels can be used as biomaterials to design and develop biomedical devices for the controlled release of drugs or phytotherapeutics, further structure-bioactivity of this type of PVA-XG hydrogel loaded with various bioactive essential oils will be investigated in our future work.

## Figures and Tables

**Figure 1 materials-15-02657-f001:**
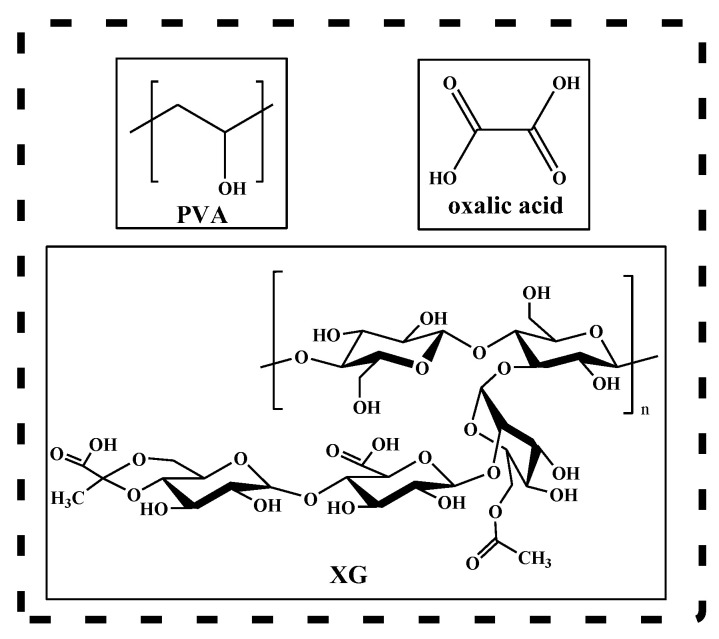
Chemical structure of PVA, XG, and oxalic acid.

**Figure 2 materials-15-02657-f002:**
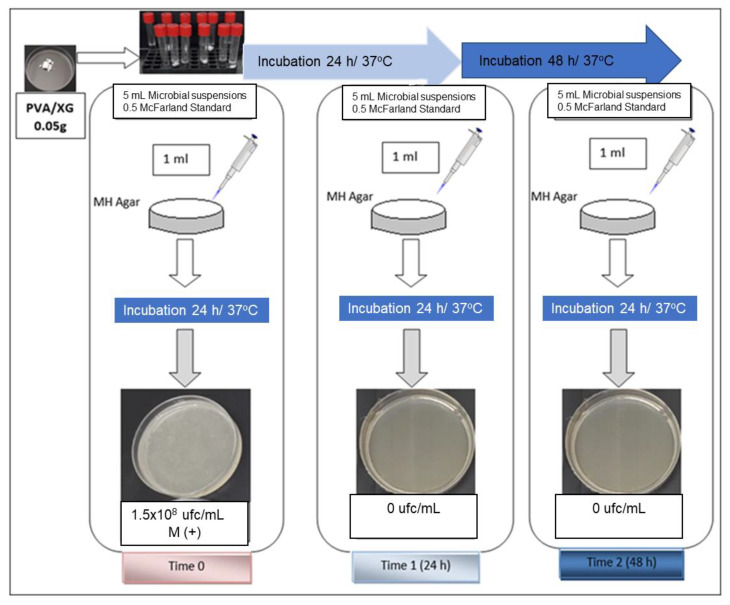
Working diagram for contact time (TC) technique: Time 0, Time 1 (24 h/37 °C), Time 2 (48 h/37 °C).

**Figure 3 materials-15-02657-f003:**
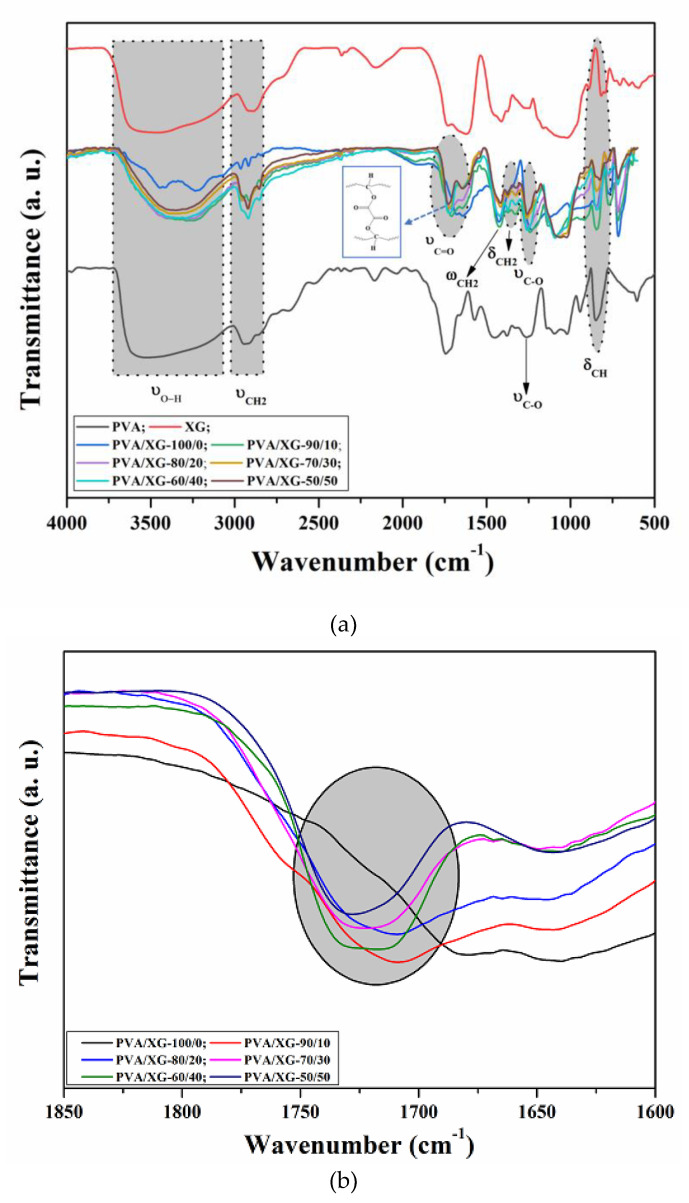
(**a**): FTIR spectra of the PVA, XG, and PVA/XG hydrogels; (**b**): Enlarged FTIR spectra in the region of 1730 cm^−1^ for the PVA, XG, and PVA/XG hydrogels.

**Figure 4 materials-15-02657-f004:**
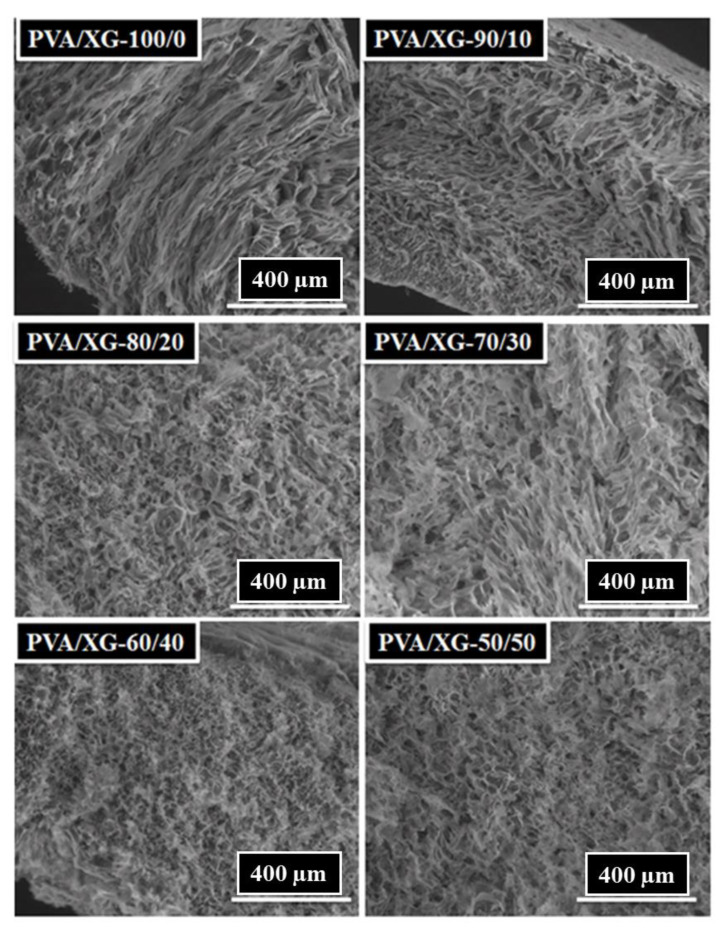
SEM images of PVA/XG hydrogels.

**Figure 5 materials-15-02657-f005:**
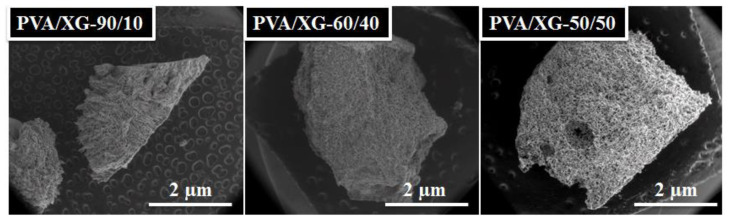
SEM images of investigated samples.

**Figure 6 materials-15-02657-f006:**
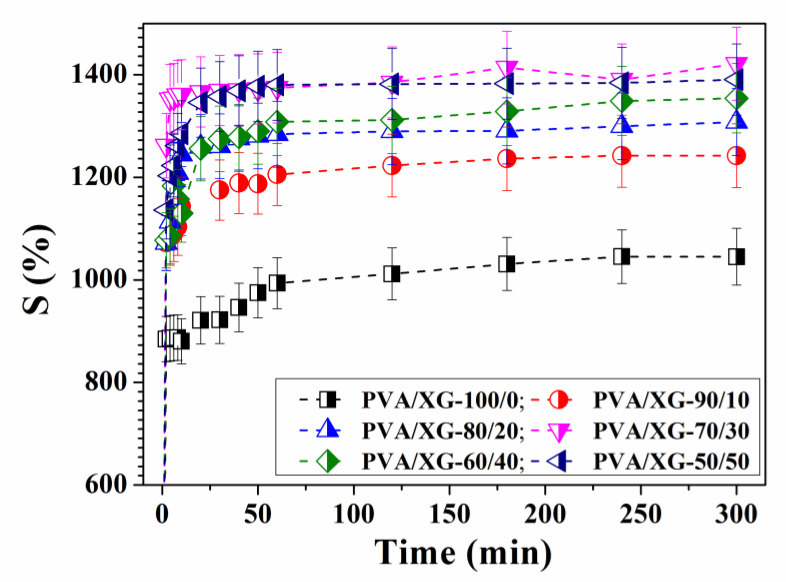
Swelling profiles in phosphate buffer pH 7.4, at 37 °C of PVA/XG systems.

**Figure 7 materials-15-02657-f007:**
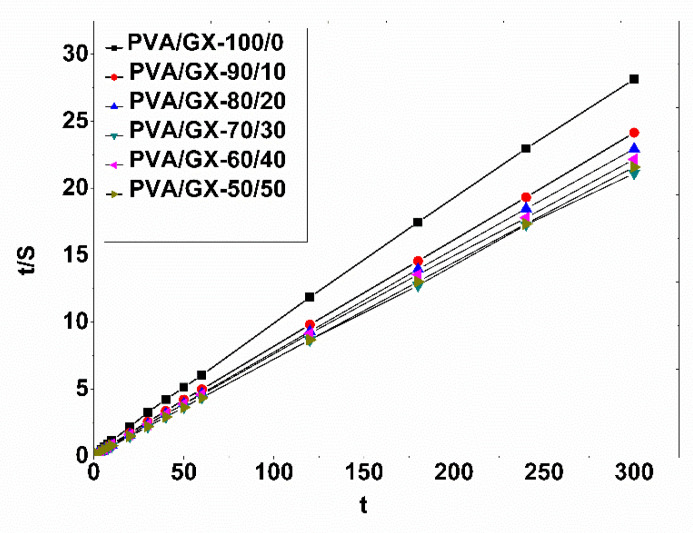
Comparison of t/S and t relationships of PVA/GX hydrogels.

**Figure 8 materials-15-02657-f008:**
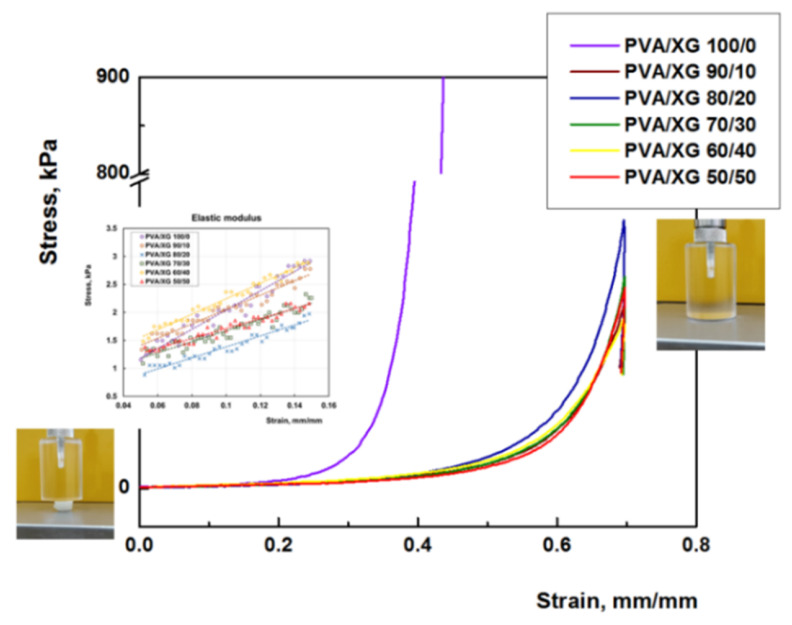
Typical stress−strain curves for hydrogels prepared with different ratios between PVA and XG; the insets present the linear dependence of stress−strain curves between 0.5 and 15% compressions, which was used to calculate the elastic modulus of all hydrogels; optical pictures of swollen PVA/XG hydrogel during the uniaxial compression test.

**Figure 9 materials-15-02657-f009:**
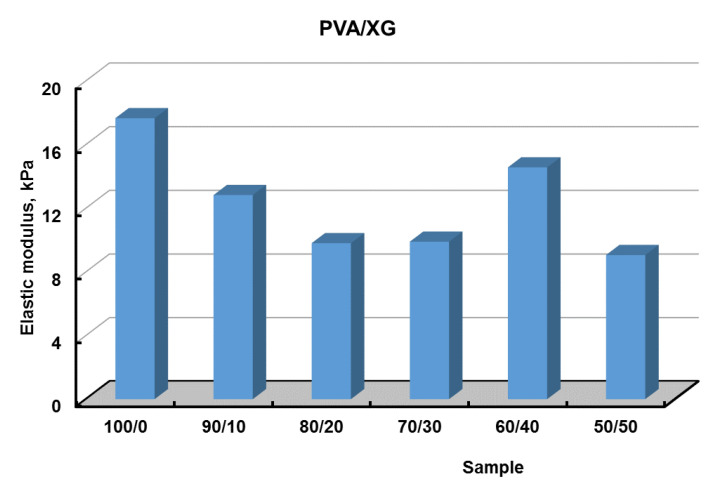
The values of elastic modulus for PVA/XG hydrogels prepared with different ratios between PVA and XG.

**Figure 10 materials-15-02657-f010:**
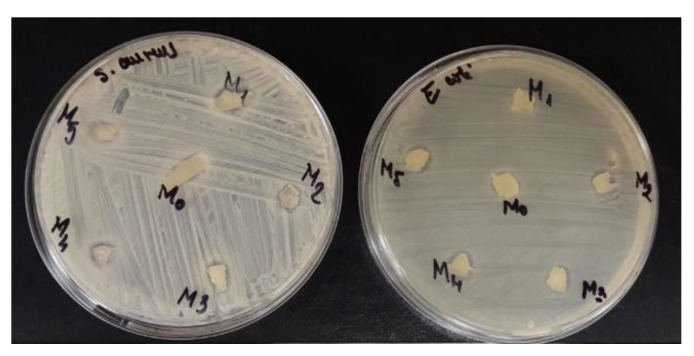
M_o_ = PVA/XG-100/0; M_1_ = PVA/XG-90/10; M_2_ = PVA/XG-80/20; M_3_ = PVA/XG-70/30; M_4_ = PVA/XG-60/40; M_5_ = PVA/XG-50/50.

**Figure 11 materials-15-02657-f011:**
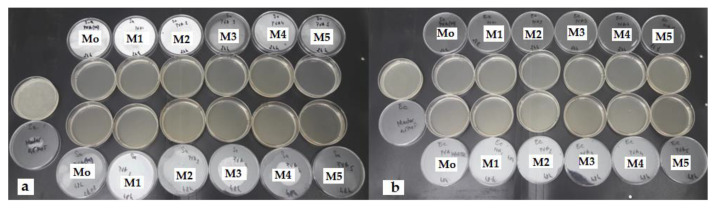
(**a**). *Staphylococcus aureus ATCC 25923*; (**b**). *Escherichia coli ATCC 25,922* M_o_ = PVA/XG-100/0; M_1_ = PVA/XG-90/10; M_2_ = PVA/XG-80/20; M_3_ = PVA/XG-70/30; M_4_ = PVA/XG-60/40; M_5_ = PVA/XG-50/50; M.

**Table 1 materials-15-02657-t001:** The composition of PVA/XG hydrogels.

Samples	PVA (g)	Xanthan Gum (g)	Oxalic Acid (g)	Distilled Water (mL)/(%)
PVA/XG-100/0	0.5	0	0.08	7.25/8
PVA/XG-90/10	0.45	0.05	0.08	7.25/8
PVA/XG-80/20	0.4	0.1	0.08	7.25/8
PVA/XG-70/30	0.35	0.15	0.08	7.25/8
PVA/XG-60/40	0.3	0.2	0.08	7.25/8
PVA/XG-50/50	0.25	0.25	0.08	7.25/8

**Table 2 materials-15-02657-t002:** Porosity, average pore diameters, equilibrium swelling ratio (S_eq_), and swelling. Kinetics parameters for PVA/XG hydrogels.

Samples	Porosity (%)	Average Pore Diameters (mm)	S_eq_ ^a^ (%)	Schott’s Second Order Kinetics		
				S_∞_ ^b^ (g/g)	K_s_^c^ (g/min *g)	R^2d^
PVA/XG-100/0	70	0.028 ± 0.007	1105.17 ± 55.25	10.81	0.02	0.9998
PVA/XG-90/10	69	0.029 ± 0.017	1242.44 ± 62.12	12.43	0.07	0.9999
PVA/XG-80/20	70	0.030 ± 0.009	1308.10 ± 65.40	13.05	0.1	0.9999
PVA/XG-70/30	70	0.040 ± 0.015	1421.28 ± 71.06	14.12	0.09	0.9999
PVA/XG-60/40	67	0.060 ± 0.011	1354.16 ± 67.70	13.53	0.04	0.9999
PVA/XG-50/50	86	0.064 ± 0.024	1390.50 ± 64.52	13.90	0.11	0.9999

^a^ equilibrium swelling ratio; ^b^ theoretical equilibrium buffer solution absorption capacity; ^c^ constant rate of swelling; ^d^ regression coefficients.

**Table 3 materials-15-02657-t003:** Results of PVA/XG hydrogel antimicrobial activity testing.

Samples	Weight (g)	*Staphylococcus aureus ATCC 25923*				*Escherichia coli ATCC 25922*			
		DM	CTM			DM	CTM		
			M(+) CFU/mL	24 h CFU/mL	48 h CFU/mL		M(+) CFU/mL	24 h CFU/mL	48 h CFU/mL
PVA/XG-100/0	0.05	+	1.5 × 10^8^	0	0	+	1.5 × 10^8^	0	0
PVA/XG-90/10	0.05	+	1.5 × 10^8^	0	0	+	1.5 × 10^8^	0	0
PVA/XG-80/20	0.05	+	1.5 × 10^8^	0	0	+	1.5 × 10^8^	0	0
PVA/XG-70/30	0.05	+	1.5 × 10^8^	0	0	+	1.5 × 10^8^	0	0
PVA/XG-60/40	0.05	+	1.5 × 10^8^	0	0	+	1.5 × 10^8^	0	0
PVA/XG-50/50	0.05	+	1.5 × 10^8^	0	0	+	1.5 × 10^8^	0	0

DM = diffusimetric method; CTM = contact time method; M (+) = positive control; CFU/mL = colony-forming unit/milliliter.

## Data Availability

The data that support the findings of this study are contained within the article.

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
