# Peer review of "Tunable Properties via Composition Modulations of Poly(vinyl alcohol)/Xanthan Gum/Oxalic Acid Hydrogels"

_materials, 2022, doi:10.3390/ma15072657_

Round 1
Reviewer 1 Report
Journal: Materials
Manuscript ID: 1663010
Manuscript title: Tunable Properties via Composition Modulations of Poly(vinyl alcohol)/Xanthan Gum/Oxalic Acid Hydrogels
The manuscript reports the preparation of cross-linked Poly(vinyl alcohol)/ xanthan gum in the presence of oxalic acid by freeze/thaw cycles method. for application as antibacterial activity. The authors have characterized polymer, they have measured some of the physical, mechanical, and biological properties. The work objectives, experimental, results, and discussion clarity are acceptable.
Generally, the paper is well written and the experimental techniques and results are thoroughly presented and discussed.
I suggest accepting this manuscript for publication after minor revision
Comments:
- Figure 2. Please delete the red wavy line in (ufc/ml)
- Figure 3. FTIR spectra, please replace it with a clearer image.
- “From Table 2 it can be seen that the porosity of PVA/XG hydrogels varies in the range of 67-86%”. (Corrected range is (70-86%))
- Authors may add DSC thermogram of PVA pure, PVA/XG and PVA/XG/oxalic acid.
Author Response
Reply to the comments of Reviewer 1:
Reviewer 1
Thank you very much for reviewing our manuscript. We carefully revised the manuscript according to your valuable comments. Our point-by-point replies to the comments of the reviewer and the yellowed changed parts in the revised manuscript were specified below:
Comments:
Q1: Figure 2. Please delete the red wavy line in (ufc/ml).
A1: The red line in (ufc/ml) was deleted and the figure 2 was replaced in the manuscript with the new one.
Q2: Figure 3. FTIR spectra, please replace it with a clearer image.
A2: The FTIR spectra was replaced with the new one prepared according to the Journal requirements (300x300 dpi).
Q3: “From Table 2 it can be seen that the porosity of PVA/XG hydrogels varies in the range of 67-86%”. (Corrected range is (70-86%)).
A3: The range provided in the initial manuscript is correct. If we look in the Table 2 we can observe that the sample denoted as PVA/GX-60/40 show a porosity equal with 67%.
Q4: Authors may add DSC thermogram of PVA pure, PVA/XG and PVA/XG/oxalic acid.
A4: Unfortunately, it is impossible for us, at this moment to add the DSC thermograms for PVA, PVA/XG and PVA/XG/oxalic acid because we don’t have the possibility to perform this analysis.

Reviewer 2 Report
The paper reports about the characterization and biological properties of PVA-Xanthan hydrogels cross-linked with oxalic acid. The system is interesting (although not very new) and the paper is recommended for publication after the following points are addressed.
Page 2. In the introduction the aim the work should be more clearly stated. Moreover, the last part (from line 70 to the end) is not suitable for an introduction.
Page 3, line 102. The authors should report how the samples were prepared for the SEM measurements. Were the samples graphitized or metallized? The mode used to collect the images (secondary or backscattered electrons) must be reported.
Pages 6 and 7 and figure 3. The evidence for the crosslinking with oxalic acid is provided only by the FTIR. I would suggest to enlarge the region around 1730 cm-1 of the spectra, because in the present figure it is not so evident that the spectrum of PVA does not show a band in this wavenumber range.
Pages 7-9. Porosity. The porosity observed in the SEM images may depend on the drying of the hydrogels. For instance the porous structures can collapse during the drying. The authors should discuss this point.
Page 10, line 330. In eq. 5, t should be divided by S.
It would be useful to show together with the experimental points also the fitting curves.
There are several typos (for instance Page 5, line 173. Page 6, line 216).
Author Response
Reply to the comments of Reviewer 2:
Reviewer 2
Thank you very much for reviewing our manuscript. We carefully revised the manuscript according to your valuable comments. Our point-by-point replies to the comments of the reviewer and the yellowed changed parts in the revised manuscript were specified below:
The paper reports about the characterization and biological properties of PVA-Xanthan hydrogels cross-linked with oxalic acid. The system is interesting (although not very new) and the paper is recommended for publication after the following points are addressed.
Q1: Page 2. In the introduction the aim the work should be more clearly stated. Moreover, the last part (from line 70 to the end) is not suitable for an introduction.
A1: The aim of the current work has been clearly stated. Parts of the introduction section have been rephrased and/or improved in the revised manuscript.
Q2: Page 3, line 102. The authors should report how the samples were prepared for the SEM measurements. Were the samples graphitized or metallized? The mode used to collect the images (secondary or backscattered electrons) must be reported.
A2: . At page 3 was added the following paragraph: “For these investigation, small pieces (≈5 mm) of the PVA/XG hydrogels were attached by conductive adhesive tape to an aluminum specimen holder. Then the PVA/XG hydrogels were inserted in the SEM microscope and observed in high vacuum using a secondary electron detector at accelerating voltage equal with 20 kV.”
Q3: Pages 6 and 7 and figure 3. The evidence for the crosslinking with oxalic acid is provided only by the FTIR. I would suggest to enlarge the region around 1730 cm-1 of the spectra, because in the present figure it is not so evident that the spectrum of PVA does not show a band in this wavenumber range.
A3: Thanks for the suggestion! We have provided Figure 3b in the revised manuscript.
Q4: Pages 7-9. Porosity. The porosity observed in the SEM images may depend on the drying of the hydrogels. For instance, the porous structures can collapse during the drying. The authors should discuss this point.
A4: The reviewer has wright. The SEM images presented in the manuscript are taken for the samples that have been thermally treated for 1h at 100oC. Indeed, on continuing applying the thermal treatment for an additional hour a collapse of the porous structure was observed as can be seen in the following images provided only for the reviewer attention.
PVA/XG – 90/10, 1h |
PVA/XG – 90/10, 2h |
PVA/XG – 50/50, 1h |
PVA/XG – 50/50, 2h |
Q5: Page 10, line 330. In eq. 5, t should be divided by S. It would be useful to show together with the experimental points also the fitting curves.
A5: The correction was made in the revised manuscript. The following figure was added in the manuscript.
Figure 7. Comparison of t/S and t relationships of PVA/GX hydrogels.
The linear relationships between t and t/S were represented in the graphs of Figure 7.
Q6: There are several typos (f
Reply to the comments of Reviewer 2:
Reviewer 2
Thank you very much for reviewing our manuscript. We carefully revised the manuscript according to your valuable comments. Our point-by-point replies to the comments of the reviewer and the yellowed changed parts in the revised manuscript were specified below:
The paper reports about the characterization and biological properties of PVA-Xanthan hydrogels cross-linked with oxalic acid. The system is interesting (although not very new) and the paper is recommended for publication after the following points are addressed.
Q1: Page 2. In the introduction the aim the work should be more clearly stated. Moreover, the last part (from line 70 to the end) is not suitable for an introduction.
A1: The aim of the current work has been clearly stated. Parts of the introduction section have been rephrased and/or improved in the revised manuscript.
Q2: Page 3, line 102. The authors should report how the samples were prepared for the SEM measurements. Were the samples graphitized or metallized? The mode used to collect the images (secondary or backscattered electrons) must be reported.
A2: . At page 3 was added the following paragraph: “For these investigation, small pieces (≈5 mm) of the PVA/XG hydrogels were attached by conductive adhesive tape to an aluminum specimen holder. Then the PVA/XG hydrogels were inserted in the SEM microscope and observed in high vacuum using a secondary electron detector at accelerating voltage equal with 20 kV.”
Q3: Pages 6 and 7 and figure 3. The evidence for the crosslinking with oxalic acid is provided only by the FTIR. I would suggest to enlarge the region around 1730 cm-1 of the spectra, because in the present figure it is not so evident that the spectrum of PVA does not show a band in this wavenumber range.
A3: Thanks for the suggestion! We have provided Figure 3b in the revised manuscript.
Q4: Pages 7-9. Porosity. The porosity observed in the SEM images may depend on the drying of the hydrogels. For instance, the porous structures can collapse during the drying. The authors should discuss this point.
A4: The reviewer has wright. The SEM images presented in the manuscript are taken for the samples that have been thermally treated for 1h at 100oC. Indeed, on continuing applying the thermal treatment for an additional hour a collapse of the porous structure was observed as can be seen in the following images provided only for the reviewer attention.
PVA/XG – 90/10, 1h |
PVA/XG – 90/10, 2h |
PVA/XG – 50/50, 1h |
PVA/XG – 50/50, 2h |
Q5: Page 10, line 330. In eq. 5, t should be divided by S. It would be useful to show together with the experimental points also the fitting curves.
A5: The correction was made in the revised manuscript. The following figure was added in the manuscript.
Figure 7. Comparison of t/S and t relationships of PVA/GX hydrogels.
The linear relationships between t and t/S were represented in the graphs of Figure 7.
Q6: There are several typos (for instance Page 5, line 173. Page 6, line 216).
A6: The corrections have been made in the revised manuscript.
or instance Page 5, line 173. Page 6, line 216).
A6: The corrections have been made in the revised manuscript.

Reviewer 3 Report
The paper by Enache et al. describes the preparation of a new hydrogel material based on PVA-XG, crosslinked with oxalic acid. Synergistic effect of both polymers on gel properties has been described. The conducted research is well presented and well planned. The obtained materials could be used as biocarriers of therapeutic substances.
I recommend accepting this paper for publication in Materials.
Author Response
Reply to the comments of Reviewer 3:
Reviewer 3
Thank you very much for reviewing our manuscript.
